# Adversarial Learning for Semi-Supervised Semantic Segmentation

## Abstract

We propose a method for semi-supervised semantic segmentation using the adversarial network. While most existing discriminators are trained to classify input images as real or fake on the image level, we design a discriminator in a fully convolutional manner to differentiate the predicted probability maps from the ground truth segmentation distribution with the consideration of the spatial resolution. We show that the proposed discriminator can be used to improve the performance on semantic segmentation by coupling the adversarial loss with the standard cross entropy loss on the segmentation network. In addition, the fully convolutional discriminator enables the semi-supervised learning through discovering the trustworthy regions in prediction results of unlabeled images, providing additional supervisory signals. In contrast to existing methods that utilize weakly-labeled images, our method leverages unlabeled images without any annotation to enhance the segmentation model. Experimental results on both the PASCAL VOC 2012 dataset and the Cityscapes dataset demonstrate the effectiveness of our algorithm.

## 1 Introduction

Semantic segmentation is the task to assign a semantic label, e.g., person, dog, or road, to each pixel in images. It is essential to a wide range of applications, such as autonomous driving and image editing. For decades, many methods have been proposed to tackle this task (Long et al., 2015; Zheng et al., 2015; Liu et al., 2015; Yu & Koltun, 2016; Lin et al., 2016), and abundant standard benchmark datasets have been constructed (Everingham et al., 2010; Mottaghi et al., 2014; Cordts et al., 2016; Zhou et al., 2017), targeting different sets of scene/object categories as well as various real-world applications. However, this task remains challenging because of the object/scene appearance variations, occlusions, and the lack of context understanding. Recently, Convolutional Neural Network (CNN) based methods such as fully convolutional neural network (FCN) (Long et al., 2015) have achieved significant improvement on the task of semantic segmentation, and most state-of-the-art algorithms are based on FCN with advanced modifications and additional modules.

Although CNN-based approaches have achieved astonishing performance, they require an enormous amount of training data. Different from image classification and object detection, semantic segmentation requires accurate per-pixel annotations for each training image, which can cost considerable expense and time. To ease the effort of acquiring high-quality data, semi/weakly-supervised methods have been applied to the task of semantic segmentation. These methods often assume that there is limited or none per-pixel annotations available, such as additional annotations on the image-level (Pinheiro & Collobert, 2015; Papandreou et al., 2015; Hong et al., 2015; Qi et al., 2016; Pathak et al., 2015a), box-level (Dai et al., 2015), or point-level (Bearman et al., 2016).

In this paper, we propose a semi-supervised semantic segmentation algorithm via adversarial learning. The recent success of Generative Adversarial Networks (GANs) (Goodfellow et al., 2014) enables many possibilities for unsupervised and semi-supervised learning. A typical GAN consists of two sub-networks, i.e., generator and discriminator, in which these two sub-networks play a min-max game in the training process. The generator takes a sample vector and outputs a sample of the target data distribution, e.g., human faces, while the discriminator aims to differentiate generated samples from target ones. Then the generator is trained to confuse the discriminator through back-propagation and therefore generates samples that are similar to the target distribution. In this paper, we apply a similar methodology and treat the segmentation network as the generator in a GAN framework.

Different from the typical generators that are trained to generate images given noise vectors, our segmentation network outputs the probability maps of the semantic labels given an input image. Under this setting, we wish to push the outputs of the segmentation network as close as the ground truth label maps spatially.

To this end, we adopt an adversarial learning scheme and propose a fully convolutional discriminator that learns to differentiate ground truth label maps from probability maps of segmentation predictions. Combined with the spatial cross-entropy loss, our method use an adversarial loss that encourages the segmentation network to produce predicted probability maps close to the ground truth label maps in a high-order structure. The idea is similar to the use of probabilistic graphical models such as Conditional Random Fields (CRFs) (Zheng et al., 2015; Chen et al., 2017; Lin et al., 2016), but without the extra post-processing module during the testing phase. In addition, the discriminator is not required during inference, and hence our proposed framework does not increase any computational power for testing. By employing the adversarial learning, we further take advantage of the proposed fully convolutional discriminator under the semi-supervised setting.

One way to allow the discriminator exploiting unlabeled data is to train the segmentation network using the adversarial learning without the cross-entropy loss. However, this approach does not improve the performance according to our experiments because the adversarial loss will aggressively encourage the predictions to be close to the ground truth distribution and neglects the correctness of segmentation. Instead, we utilize the confidence maps generated by our discriminator network as the supervisory signal to guide the cross-entropy loss in a "self-taught" manner. The confidence maps indicate which regions of the prediction distribution are close to the ground truth label distribution, so that the segmentation network can trust these predictions and hence can be trained via a masked cross-entropy loss. By adopting the proposed framework, we show that the segmentation accuracy can be further improved by adding images without any annotations in the domain of labeled images.

The contributions of this work are as follows. First, we develop an adversarial framework that improves semantic segmentation accuracy without requiring additional computation loads during inference. Second, we facilitate the semi-supervised learning by leveraging the discriminator network response of unlabeled images to aid the training of the segmentation network. Experimental results validate the proposed adversarial framework for semi-supervised semantic segmentation on the PASCAL VOC 2012 (Everingham et al., 2010) and Cityscapes (Cordts et al., 2016) datasets.

## 2    RELATED WORK

**Semantic segmentation.** Recent state-of-the-art methods for semantic segmentation are based on the rapid development of CNN. As proposed by Long et al. (2015), one can transform a classification CNN, e.g. AlexNet (Krizhevsky et al., 2012), VGG (Simonyan & Zisserman, 2015), or ResNet (He et al., 2016), to a fully-convolutional network (FCN) that tackles the task of semantic segmentation. However, pixel-level annotations are usually expensive and difficult to collect. To reduce the heavy effort of labeling segmentation ground truth, many weakly-supervised approaches are proposed in recent years. In the weakly-supervised setting, the segmentation network is not trained at the pixel level with fully annotated ground truth. Instead, the network is trained with various weak-supervisory signals that are more easily to obtain. Image-level labels are exploited as the supervisory signal in most methods. Pinheiro & Collobert (2015) and Pathak et al. (2015b) use Multiple Instance Learning (MIL) to generate latent segmentation label maps for supervised training. On the other hand, Papandreou et al. (2015) refer to the image-level labels to penalize the prediction of non-existent object classes, while similarly Qi et al. (2016) use object localization to refine the segmentation. Hong et al. (2015) refer to the labeled images to train a classification network as the feature extractor for deconvolution. In addition to image-level supervisions, the segmentation network can also be trained with bounding boxes (Dai et al., 2015; Khoreva et al., 2017), point supervision (Bearman et al., 2016), or web videos (Hong et al., 2017).

However, these weakly supervised approaches still fall behind the fully-supervised ones, especially because the detailed boundary information is difficult to infer from these weak-supervisory signals. Hence semi-supervised learning is also considered in some methods to enhance the prediction performance. In such setting, partial fully-annotated data and optional weakly-labeled data are used for the segmentation network training. Hong et al. (2015) jointly train their network with image-level supervised images and few fully-annotated images in the encoder-decoder framework.

Dai et al. (2015) and Papandreou et al. (2015) also expand their weakly-supervised approaches to the semi-supervised setting for utilizing extra strongly-annotated data.

Different from the aforementioned methods, our method can leverage unlabeled images in model training, hence greatly saving the cost of manual annotation. In fact, we treat the output of our fully convolutional discriminator as the supervisory signals, which compensate for the absence of image annotations and enable semi-supervised semantic segmentation. Our self-taught learning framework for segmentation is related to Pathak et al. (2015a) where the prediction maps of unlabeled images are used as ground truth. However, in Pathak et al. (2015a), the prediction maps are refined by several hand-designed constraints before training, while we learn the confidence map through the discriminator network as the selection criterion for self-taught learning.

**Generative adversarial networks.** After Goodfellow et al. (2014) propose the GAN framework and its theoretical foundation, the GAN draws great attention with several improvements in implementation (Radford et al., 2016; Denton et al., 2015; Arjovsky et al., 2017; Mao et al., 2016; Berthelot et al., 2017). The methodology of adversarial training has been applied to a wide range of applications, including image genration (Radford et al., 2016), image completion (Li et al., 2017), super-resolution (Ledig et al., 2016), object detection (Wang et al., 2017), domain adaptation (Hoffman et al., 2016) and semantic segmentation (Luc et al., 2016; Souly et al., 2017).

The work closest in scope to ours is the one proposed by Luc et al. (2016), where the adversarial network is used to aid the training for semantic segmentation. However, it does not show substantial improvement over the baseline. On the other hand, Souly et al. (2017) propose to generate adversarial examples using GAN for semi-supervised semantic segmentation, but these generated examples may not be sufficiently close to real images to help the segmentation network.

## 3 ALGORITHM OVERVIEW

Figure 1 shows the overview of the proposed algorithm. Our system is composed of two networks: the segmentation network and the discriminator network. The former can be any network designed for semantic segmentation, e.g., FCN (Long et al., 2015), DeepLab (Chen et al., 2017), DilatedNet (Yu & Koltun, 2016). Given an input image with dimension $H \times W \times 3$, the segmentation network outputs the class probability maps of size $H \times W \times C$, where $C$ is the number of semantic categories of the target dataset.

Our discriminator network is an FCN-based network, which takes class probability maps as the input, either from the segmentation network or ground truth label maps, and then outputs spatial probability maps with a size of $H \times W \times 1$. Each pixel of the discriminator outputs map represents whether that pixel is sampled from the ground truth label ($p = 1$) or from the segmentation network ($p = 0$). In contrast to the typical GAN discriminators which take fix-sized input images ($64 \times 64$ in most cases) and output a single probability value, we transform our discriminator to a fully-convolutional network that can take inputs of arbitrary sizes. Importantly, we find this transformation is essential to enable the proposed adversarial learning scheme.

During the training process, we use both labeled and unlabeled images under the semi-supervised setting. When using the labeled data, the segmentation network is supervised by both the standard cross-entropy loss with the ground truth label map and the adversarial loss with the discriminator network. Note that we train the discriminator network only with the labeled data.

For the unlabeled data, we train the segmentation network with the proposed semi-supervised method. After obtaining the initial segmentation prediction of the unlabeled image from the segmentation network, we obtain a confidence map by passing the segmentation prediction through the discriminator network. We in turn treat this confidence map as the supervisory signal using a "self-taught" scheme to train the segmentation network with a masked cross-entropy loss. The intuition is that this confidence map indicates the local quality of the predicted segmentation, so that the segmentation network knows which regions to trust during training.

## 4 SEMI-SUPERVISED TRAINING WITH ADVERSARIAL NETWORK

In this section, we address the detailed learning scheme of the segmentation and discriminator networks, as well as the designed network architectures.

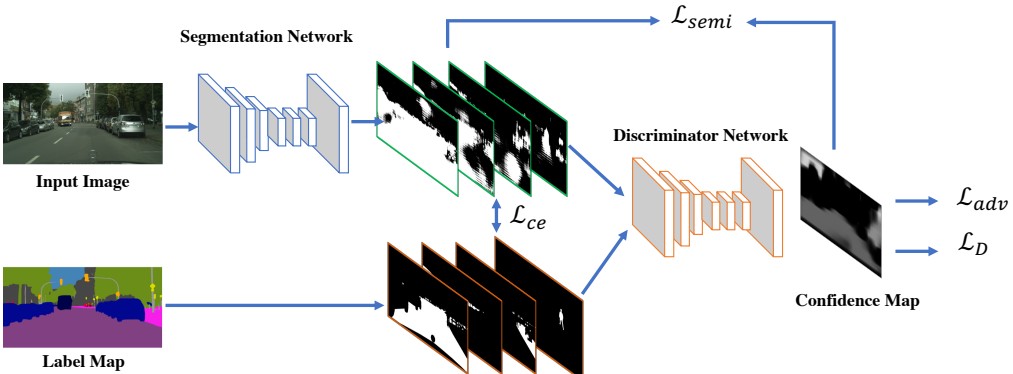

Figure 1: Overview of the proposed system for semi-supervised semantic segmentation. With a fully-convolution discriminator network trained using the loss $\mathcal{L}_D$, we optimize the segmentation netwrok using three loss functions during the training process: cross-entropy loss $\mathcal{L}_{ce}$, adversarial loss $\mathcal{L}_{adv}$, and semi-supervised loss $\mathcal{L}_{semi}$.

## 4.1 TRAINING OBJECTIVE

Given an input image $\mathbf{X}_n$ of size $H \times W \times 3$, we denote the segmentation network as $S(\cdot)$ and the predicted probability map as $S(\mathbf{X}_n)$ of size $H \times W \times C$, where C is the category number. For our fully convolutional discriminator, we denote it as $D(\cdot)$ which outputs a two-class confidence map $D(\mathbf{P}_n)$ with the size of $H \times W \times 1$, where $\mathbf{P}_n$ is the class probability map of size $H \times W \times C$, from either the ground truth label $Y_n$ or the segmentation network as $S(\mathbf{X}_n)$.

**Discriminator network training.** To train the discriminator network, we minimize the spatial cross-entropy loss $\mathcal{L}_D$ with respect to two classes. The loss can be formally written as:

$$\mathcal{L}_D = -\sum_{h,w} (1 - y_n) \log(D(\mathbf{P}_n)^{(h,w,0)}) + y_n \log(D(\mathbf{P}_n)^{(h,w,1)}), \tag{1}$$

where $y_n = 0$ if the sample is drawn from the segmentation network, and $y_n = 1$ if the sample is from the ground truth label. Note that, the discriminator network takes a C-channel probability map as input. In order to convert the ground truth label map $Y_n$ of size $H \times W \times 1$ to C channels, we simply employ one-hot encoding scheme by constructing the probability maps $P_n$, where $P_n^{(h,w,c)}$ takes value 1 if $Y_n^{(h,w)} = c$, and 0 otherwise.

One concern raised by (Luc et al., 2016) is that the discriminator network may easily distinguish whether the probability maps come from the ground truth by detecting the one-hot probability. However, we do not observe this phenomenon during the training phase. One reason is that we use a fully-convolutional scheme to predict spatial confidence, which increases the difficulty to learn the discriminator. In addition, we try the *Scale* scheme proposed in (Luc et al., 2016), where the ground truth probability channel is slightly diffused to other channels according to the distribution of segmentation network output. However, the results show no difference, and thus we do not adopt this scheme in the experiments.

**Segmentation network training.** We propose to train the segmentation network via minimizing a multi-task loss function:

$$\mathcal{L}_{seg} = \mathcal{L}_{ce} + \lambda_{adv}\mathcal{L}_{adv} + \lambda_{semi}\mathcal{L}_{semi}, \tag{2}$$

where $\mathcal{L}_{ce}$, $\mathcal{L}_{adv}$, and $\mathcal{L}_{semi}$ denote the spatial multi-class cross entropy loss, the adversarial loss, and the semi-supervised loss, respectively. $\lambda_{adv}$ and $\lambda_{semi}$ are two constants for balancing the multi-task training.

We first consider the scenario of using annotated data. Given an input image $\mathbf{X}_n$, ground truth $\mathbf{Y}_n$ and prediction results $\mathbf{P}_n = S(\mathbf{X}_n)$, the cross-entropy loss is obtained by:

$$\mathcal{L}_{ce} = -\sum_{h,w} \sum_{c \in C} \mathbf{Y}_n^{(h,w,c)} \log(\mathbf{P}_n^{(h,w,c)}). \tag{3}$$

We adopt the adversarial learning through the adversarial loss $\mathcal{L}_{adv}$ given a fully convolutional discriminator network $D(\cdot)$:

$$\mathcal{L}_{adv} = -\sum_{h,w} \log(D(\mathbf{P}_n)^{(h,w,1)}). \tag{4}$$

With this adversarial loss, we seek to train the segmentation network to fool the discriminator by maximizing the probability of the segmentation prediction being considered as the ground truth distribution.

**Training with unlabeled data.** Now we consider the adversarial training under the semi-supervised setting. For unlabeled data, it is obvious that we cannot apply $\mathcal{L}_{ce}$ since there is no ground truth annotation available. The adversarial loss $\mathcal{L}_{adv}$ is still applicable as it only requires the discriminator network. However, we find that the performance degenerates when only applying the adversarial loss on unlabeled data without $\mathcal{L}_{ce}$. This is reasonable because the discriminator serves as a regularization and may over-correct the prediction to fit the ground truth distribution.

Thus, we propose to utilize the trained discriminator with unlabeled data using a "self-taught" strategy. The main idea is that the trained discriminator can generate a confidence map, i.e. $D(\mathbf{P}_n)^{(h,w,1)}$, which infers the regions where the prediction results are close enough to the ground truth distribution. We then binarize this confidence map with a threshold to highlight the trustworthy region. As a result, we define the self-taught ground truth as the masked segmentation prediction $\hat{\mathbf{Y}}_n = argmax(\mathbf{P}_n)$ using this binarized confidence map. The resulting semi-supervised loss is defined by:

$$\mathcal{L}_{semi} = -\sum_{h,w} \sum_{c \in C} I(D(\mathbf{P}_n)^{(h,w,1)} > T_{semi}) \cdot \hat{\mathbf{Y}}_n^{(h,w,c)} \log(\mathbf{P}_n^{(h,w,c)}), \tag{5}$$

where $I(\cdot)$ is the indicator function, and $T_{semi}$ is the threshold to control the sensitivity of the self-taught process. Note that during training we treat both the self-taught target $\hat{\mathbf{Y}}_n$ and the value of indicator function as constant, and thus (5) can be simply viewed as a masked spatial cross entropy loss. In practice, we find that this strategy works robustly with $T_{semi}$ ranging between $0.1$ and $0.3$.

## 4.2 NETWORK ARCHITECTURE

**Segmentation network.** We adopt the DeepLab-v2 (Chen et al., 2017) framework with ResNet-101 (He et al., 2016) model pre-trained on the ImageNet dataset (Deng et al., 2009) as our segmentation baseline network. However, we do not employ the multi-scale fusion proposed in Chen et al. (2017) due to the memory concern. Following the practice of recent work on semantic segmentation (Chen et al., 2017; Yu & Koltun, 2016), we remove the last classification layer and modify the stride of the last two convolution layers from 2 to 1, making the resolution of the output feature maps effectively $1/8$ times the input image size. To enlarge the receptive fields, we apply the dilated convolution (Yu & Koltun, 2016) in conv4 and conv5 layers with a stride of 2 and 4, respectively. After the last layer, we employ the Atrous Spatial Pyramid Pooling (ASPP) proposed in Chen et al. (2017) as the final classifier. Finally, we apply an up-sampling layer along with the softmax output to match the size of the input image.

**Discriminator network.** For the discriminator network, we follow the structure used in Radford et al. (2016). It consists of 5 convolution layers with kernel $4 \times 4$ with channel numbers $\{64, 128, 256, 512, 1\}$ and stride of 2. Each convolution layer is followed by a Leaky-ReLU (Maas et al., 2013) parameterized by $0.2$ except the last layer. To transform the network to a fully convolutional network, an up-sampling layer is added to the last layer to rescale the output to the size of the input map. Note that we do not employ the batch-normalization layers. We find that the batch-normalization layer (Ioffe & Szegedy, 2015) is highly unstable since the system can be only trained with a small batch size.

## 5 EXPERIMENTAL RESULTS

**Implementation details.** We implement our network using the PyTorch framework. We train our system on a single TitanX GPU with 12 GB memory. To train the segmentation network, we use

Table 1: Results on the VOC 2012 *validation* set.

| Methods | Data Amount | | | |
|---|---|---|---|---|
| | 1/8 | 1/4 | 1/2 | Full |
| FCN-8s (Long et al., 2015) | N/A | N/A | N/A | 67.2 |
| Dilation10 (Yu & Koltun, 2016) | N/A | N/A | N/A | 73.9 |
| DeepLab-v2 (Chen et al., 2017) | N/A | N/A | N/A | 77.7 |
| our baseline | 66.0 | 68.3 | 69.8 | 73.6 |
| baseline + $\mathcal{L}_{adv}$ | 67.6 | 71.0 | 72.6 | 74.9 |
| baseline + $\mathcal{L}_{adv}$ + $\mathcal{L}_{semi}$ | 68.8 | 71.6 | 73.2 | N/A |

Table 2: Results on the Cityscapes *validation* set.

| Methods | Data Amount | | | |
|---|---|---|---|---|
| | 1/8 | 1/4 | 1/2 | Full |
| FCN-8s (Long et al., 2015) | N/A | N/A | N/A | 65.3 |
| Dilation10 (Yu & Koltun, 2016) | N/A | N/A | N/A | 67.1 |
| DeepLab-v2 (Chen et al., 2017) | N/A | N/A | N/A | 70.4 |
| our baseline | 52.4 | 58.3 | 62.6 | 66.4 |
| baseline + $\mathcal{L}_{adv}$ | 53.8 | 59.1 | 63.7 | 67.7 |
| baseline + $\mathcal{L}_{adv}$ + $\mathcal{L}_{semi}$ | 54.2 | 59.7 | 64.5 | N/A |

Stochastic Gradient Descent (SGD) with Nesterov acceleration as the optimizer, where the momentum is 0.9 and the weight decay is $10^{-4}$. The initial learning rate is set as $2.5 \times 10^{-4}$ and is decreased with polynomial decay with power of 0.9 as mentioned in Chen et al. (2017). For training the discriminator, we adopt Adam optimizer (Kingma & Ba, 2014) with the learning rate as $10^{-4}$ and the same polynomial decay as the segmentation network. The momentum is set as 0.9 and 0.999.

For semi-supervised training, we randomly interleave the labeled data and unlabeled data iteratively and apply the training scheme described in section 4.1 accordingly. We update both the segmentation network and discriminator network jointly. In each iteration, only the batch containing the ground truth data are used for training the discriminator. When randomly sampling partial labeled and unlabeled data from the datasets, we average several experiment results with different random seeds to ensure the evaluation robustness.

**Evaluation datasets and metric.** In this work, we conduct experiments on two semantic segmentation datasets: PASCAL VOC 2012 (Everingham et al., 2010) and Cityscapes (Cordts et al., 2016). While the PASCAL VOC dataset contains common objects in photos captured in daily activities, the Cityscapes dataset mainly targets urban street scenes. On both datasets, we use the mean intersection-over-union (mean IU) as the evaluation metric.

The PASCAL VOC 2012 dataset is a commonly used evaluation dataset for semantic segmentation. It comprises 20 common objects with annotations on daily captured photos. We use the extra annotation set in SBD (Hariharan et al., 2011), resulting in 10,582 training images. We evaluate our models on the standard validation set with 1449 images. During training, we employ the random scaling and cropping with size $321 \times 321$. We train each model on the PASCAL VOC dataset for 20k iterations with batch size 10.

The Cityscapes dataset has 50 videos with driving scenes, and 2975, 500, 1525 images are extracted and annotated with 19 classes for training, validation, and testing, respectively. Each annotated frame is the $20^{th}$ frame in a 30-frames snippet, where only these images with annotations are considered in the training process. We resize the input image to $512 \times 1024$ without any random cropping/scaling. We train each model on the Cityscapes dataset for 40k iterations with batch size 2.

**Results on the PASCAL VOC 2012 dataset.** Table 1 shows the evaluation results on the PASCAL VOC 2012 dataset. To validate the semi-supervising scheme, we randomly sample 1/8, 1/4, 1/2 images as labeled, and used the rest of training images as the unlabeled data. We show the performance comparisons with FCN (Long et al., 2015), Dilation10 (Yu & Koltun, 2016), and DeepLab-v2 (Chen et al., 2017) to demonstrate that our baseline model is comparable with other state-of-the-art methods. Note that our baseline model is equivalent to the DeepLab-v2 model without multi-scale fusion. The adversarial loss brings consistent performance improvement ($1.6\% - 2.8\%$) over different amounts of training data. Incorporating the proposed semi-supervised learning scheme brings overall $2.8\% - 3.4\%$ improvement. Figure 2 shows visual comparisons of the segmentation results generated by the proposed method. We observe that the segmentation boundary has significant improvement when compared to the baseline model.

**Results on the Cityscapes dataset.** Table 2 shows evaluation results on the Cityscapes dataset. By applying the adversarial loss $\mathcal{L}_{adv}$, the model achieves $0.8\% - 1.4\%$ gain over the baseline model under the semi-supervised setting. This shows that our adversarial training scheme can encourage the segmentation network to learn the structural information from the ground truth distribution. Combining the adversarial learning and proposed semi-supervised learning, the performance further improves with overall $1.4\% - 1.9\%$ mean IU gain.

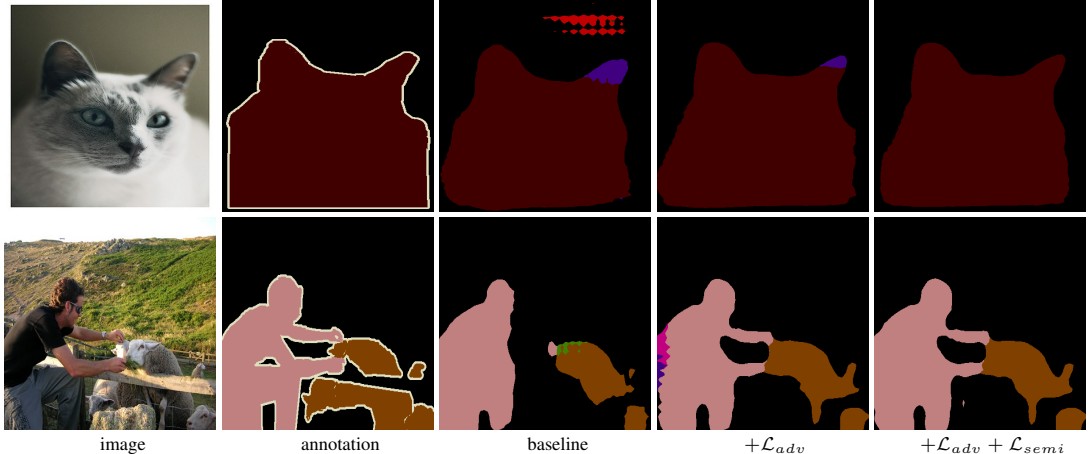

| image | annotation | baseline | $+\mathcal{L}_{adv}$ | $+\mathcal{L}_{adv} + \mathcal{L}_{semi}$ |

Figure 2: Comparisons on the PASCAL VOC 2012 dataset using 1/2 labeled data.

Table 3: Adversarial learning comparison with Luc et al. (2016) on VOC 2012 *validation* set.

|  | Baseline | Adversarial |
|---|---|---|
| Luc et al. (2016) | 71.8 | 72.0 |
| ours | 73.6 | 74.9 |

Table 4: Semi-supervised learning comparisons on VOC 2012 *validation* set without using additional labels of SBD.

|  | Data Amount | Fully-supervised | Semi-supervised |
|---|---|---|---|
| Papandreou et al. (2015) | Full | 62.5 | 64.6 |
| Souly et al. (2017) | Full | 59.5 | 64.1 |
| ours | Full | 66.3 | 68.4 |
| Souly et al. (2017) | 30% | 38.9 | 42.2 |
| ours | 30% | 57.4 | 60.6 |

**Comparisons with state-of-the-art methods** Table 3 shows comparisons with Luc et al. (2016) that utilizes adversarial learning. There are major design differences of the adversarial learning step between Luc et al. (2016) and our method. First, we design a universal discriminator for various datasets, while Luc et al. (2016) utilizes different network structures for different datasets. Second, our discriminator is not required to take the RGB image as an additional input but directly work on the prediction map from the segmentation network. In Table 3, our method achieves 1.2% gain in mean IU, which is significantly better then the gain in Luc et al. (2016).

We show comparisons for the semi-supervised setting in Table 4. To compare with Papandreou et al. (2015) and Souly et al. (2017), we train our model on the original PASCAL VOC 2012 train set (1464 images) and use the SBD (Hariharan et al., 2011) set as unlabeled data. It is worth noting that in Papandreou et al. (2015), image-level labels are available for the SBD (Hariharan et al., 2011) set, and in Souly et al. (2017), additional unlabeled images are generated through their generator during the training stage.

**Hyper-parameter analysis.** The proposed algorithm is parametrized by three hyper parameters: $\lambda_{adv}$ and $\lambda_{semi}$ are two parameters for balancing the multi-task learning in (2), and $T_{semi}$ is used to control the sensitivity in the semi-supervised learning described in (5). We evaluate these hyper parameters using the PASCAL VOC dataset under the fully/semi-supervised setting. We show comparison results of different parameter settings in Table 5. We first evaluate the effect on $\lambda_{adv}$ using fully-supervised setting. Note that we do not use any unlabeled data, i.e. $\lambda_{semi} = 0$. The baseline model without adversarial learning ($\lambda_{adv} = 0$) achieves 73.6% mean IU. When $\lambda_{adv} = 0.01$, the model achieves 74.9% mean IU with 1.3% improvement. When $\lambda_{adv} = 0.05$, the performance deprecates to 73.0% mean IU, which indicates that the adversarial loss is too large.

Second, we show comparisons of different values of $\lambda_{semi}$ with 1/8 amount of data under the semi-supervised setting. We set $\lambda_{adv} = 0.01$ and $T_{semi} = 0.2$ for the comparisons. Overall, $\lambda_{semi} = 0.1$ achieves the best performance of 68.8% mean IU with 1.2% gain.

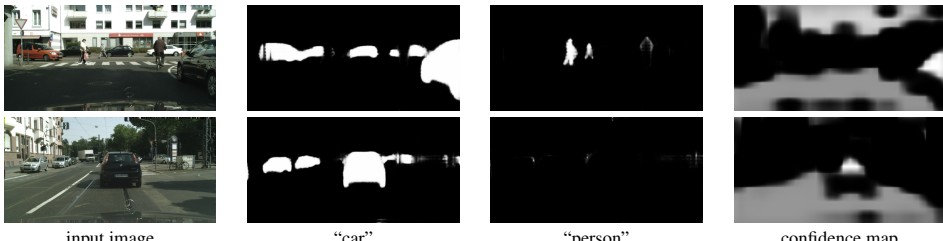

input image          "car"          "person"          confidence map

Figure 3: Visualization of the confidence maps. Given the probability maps generated by the segmentation network, the confidence maps is then obtained from the discriminator. In the confidence maps, the brighter regions indicate that they are close to the ground truth distribution.

Lastly, we perform the experiments with different value of $T_{semi}$, where we set $\lambda_{adv} = 0.01$ and $\lambda_{semi} = 0.1$ . High $T_{semi}$ suggests that we only trust regions of high structural similarity as the ground truth distribution. We find that our proposed strategy performs well for a wide range of values $T_{semi}$ (0.1 to 0.3). The method performs the best when $T_{semi} = 0.2$. When $T_{semi} = 0$, we trust all the pixel predictions in unlabeled images, resulting in performance degradation. In Figure 3, we show the visualization of generated confidence maps given the predicted probability maps.

**Ablation study.** We present the ablation study of our proposed system in Table 6 on the PASCAL VOC dataset. First, we examine the impact of using fully convolutional discriminator (FCD). To construct a discriminator that is not fully-convolutional, we replace the last convolution layer of the discriminator with a fully-connected layer that outputs a single neuron as in typical GAN models. Without using FCD, the performance drops $1.0\%$ using full data and $0.9\%$ with 1/8 data. This shows that the use of FCD is essential to the adversarial learning. Second, we apply the semi-supervised learning method without the adversarial loss. The results show that the adversarial training on the labeled data is important to our semi-supervised scheme. If the segmentation network does not seek to fool the discriminator, the confidence maps generated by the discriminator would be meaningless, providing weaker supervisory signals.

Table 5: Hyper parameter analysis.

| Data Amount | $\lambda_{adv}$ | $\lambda_{semi}$ | $T_{semi}$ | Mean IU |
|---|---|---|---|---|
| Full | 0 | 0 | N/A | 73.6 |
| Full | 0.005 | 0 | N/A | 74.0 |
| Full | 0.01 | 0 | N/A | 74.9 |
| Full | 0.02 | 0 | N/A | 74.6 |
| Full | 0.04 | 0 | N/A | 74.1 |
| Full | 0.05 | 0 | N/A | 73.0 |
| 1/8 | 0.01 | 0 | N/A | 67.6 |
| 1/8 | 0.01 | 0.05 | 0.2 | 68.6 |
| 1/8 | 0.01 | 0.1 | 0.2 | 68.8 |
| 1/8 | 0.01 | 0.2 | 0.2 | 68.5 |
| 1/8 | 0.01 | 0.1 | 0 | 66.5 |
| 1/8 | 0.01 | 0.1 | 0.1 | 68.0 |
| 1/8 | 0.01 | 0.1 | 0.2 | 68.8 |
| 1/8 | 0.01 | 0.1 | 0.3 | 68.7 |
| 1/8 | 0.01 | 0.1 | 1.0 | 67.6 |

Table 6: Ablation study of the proposed method on the PASCAL VOC dataset.

| | | | Data Amount | |
|---|---|---|---|---|
| $\mathcal{L}_{adv}$ | $\mathcal{L}_{semi}$ | FCD | 1/8 | Full |
| | | | 66.0 | 73.6 |
| ✓ | | ✓ | 67.6 | 74.9 |
| ✓ | | | 66.6 | 74.0 |
| | ✓ | ✓ | 65.7 | N/A |
| ✓ | ✓ | ✓ | 68.8 | N/A |

# 6 CONCLUSIONS

In this work, we propose an adversarial learning scheme for semi-supervised semantic segmentation. We train a fully convolutional discriminator network to enhance the segmentation network with both labeled and unlabeled data. With labeled data, the adversarial loss for the segmentation network is designed to learn higher order structural information without post-processing. For unlabeled data, the confidence maps generated by the discriminator network act as the self-taught signal for refining the segmentation network. Extensive experiments on the PASCAL VOC 2012 dataset and on the Cityscapes dataset are performed to validate the effectiveness of the proposed algorithm.

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

## A  OVERVIEW

In this appendix, we present additional results of the proposed method. First, we provide the detailed training parameters for both evaluation datasets. Second, we show more qualitative comparisons of our proposed method on both the PASCAL VOC dataset (Everingham et al., 2010) and on the Cityscapes dataset (Cordts et al., 2016).

## B  TRAINING PARAMETERS

Table 7: Training parameters.

| Parameter | Cityscaps | PASCAL VOC |
|---|---|---|
| Trained iterations | 40,000 | 20,000 |
| Learning rate | 2.5e-4 | 2.5e-4 |
| Learning rate (D) | 1e-4 | 1e-4 |
| Polynomial decay | 0.9 | 0.9 |
| Momentum | 0.9 | 0.9 |
| Optimizer | SGD | SGD |
| Optimizer (D) | Adam | Adam |
| Nesterov | True | True |
| Batch size | 2 | 10 |
| Weight decay | 0.0001 | 0.0001 |
| Crop size | 512x1024 | 321x321 |
| Random scale | No | Yes |

## C  ADDITIONAL QUALITATIVE RESULTS

In Figure 4-5, we show the additional qualitative comparisons with the models using half training data of the PSCAL VOC dataset. In Figure 6, we also show the additional qualitative comparisons with the models using half training data of the Cityscapes dataset. The results show that both the adversarial learning and the semi-supervised training scheme can improve the performance of the semantic segmentation.

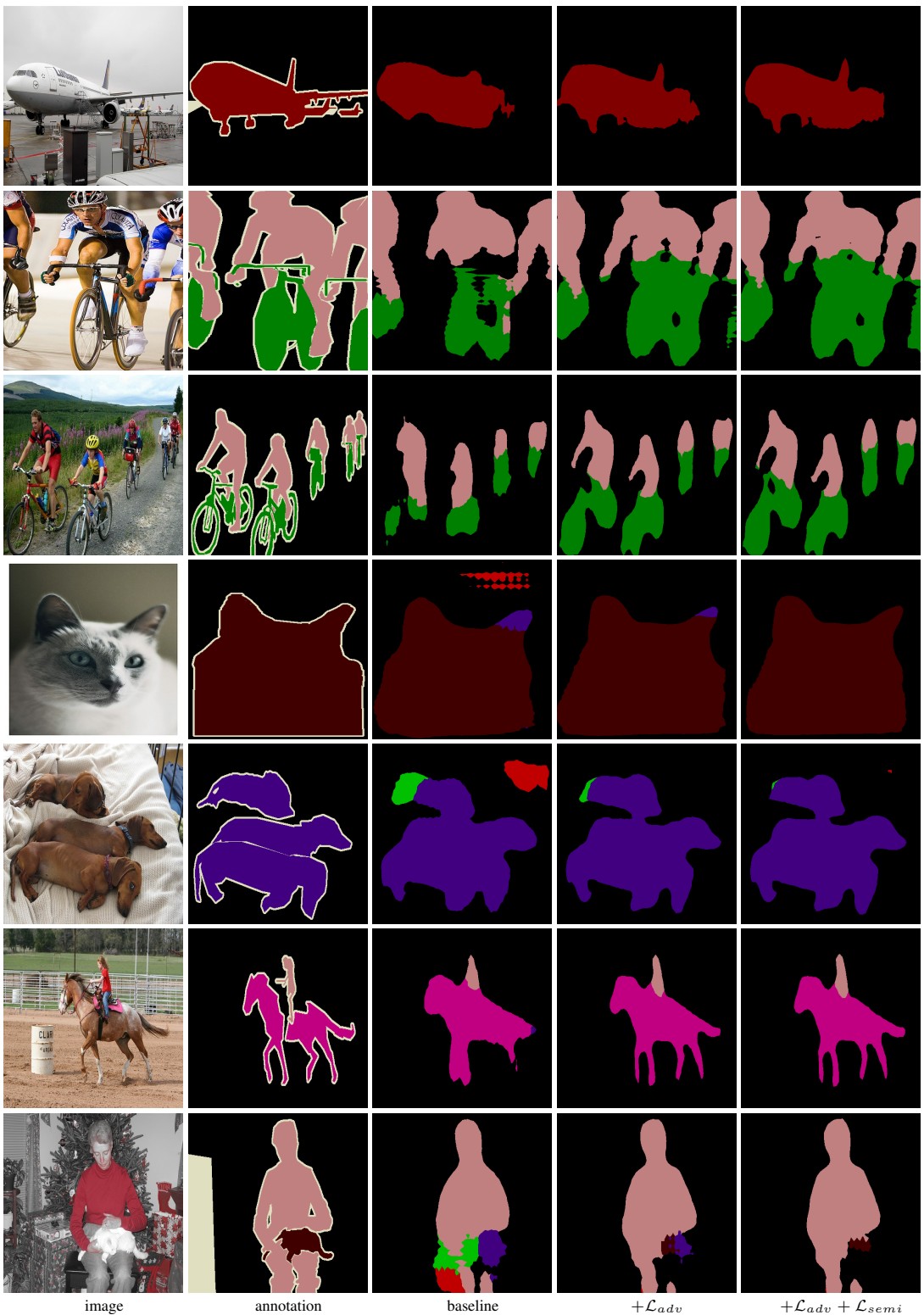

Figure 4: Comparisons on the PASCAL VOC dataset using 1/2 training data.

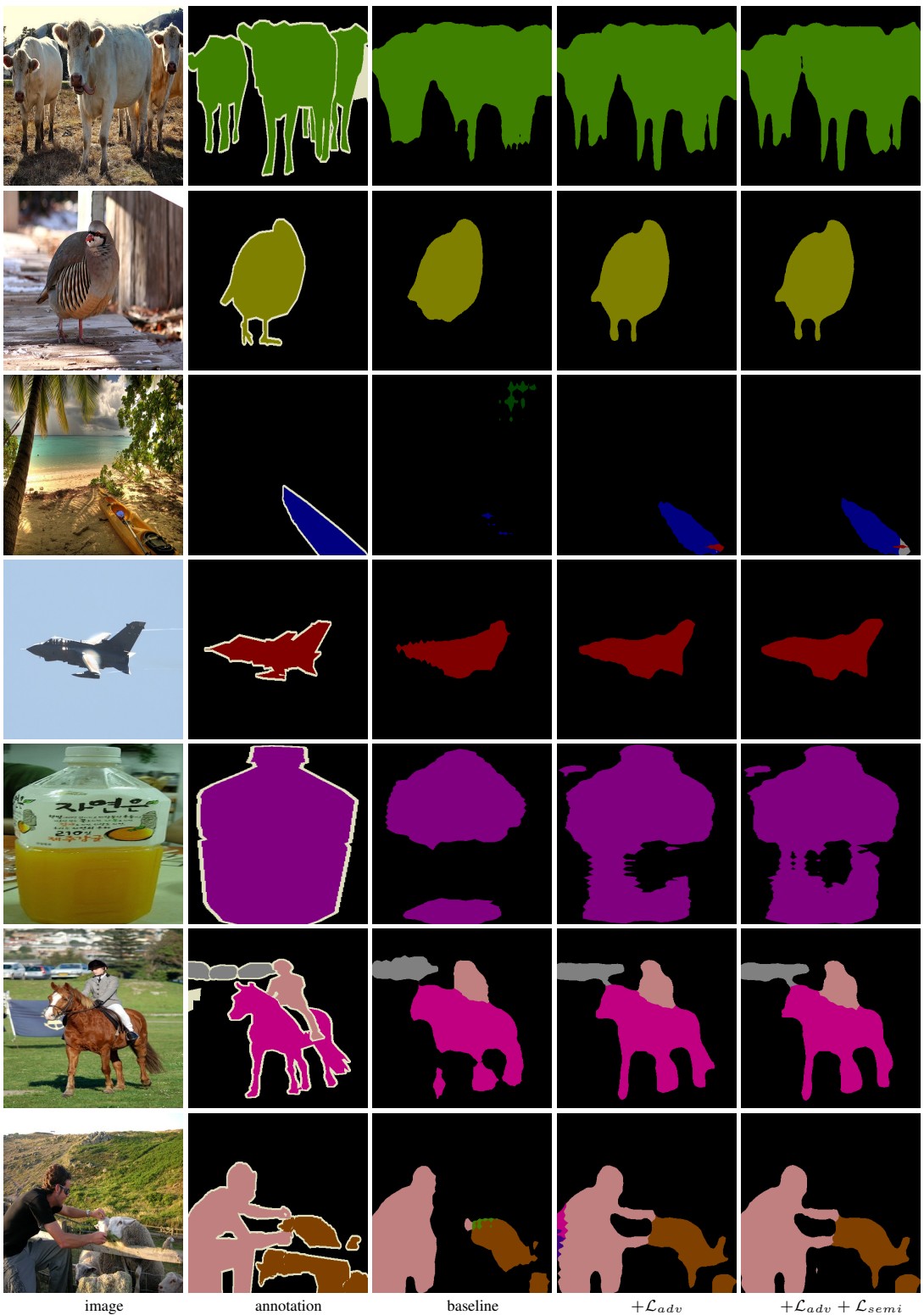

Figure 5: Comparisons on the PASCAL VOC dataset using 1/2 training data.

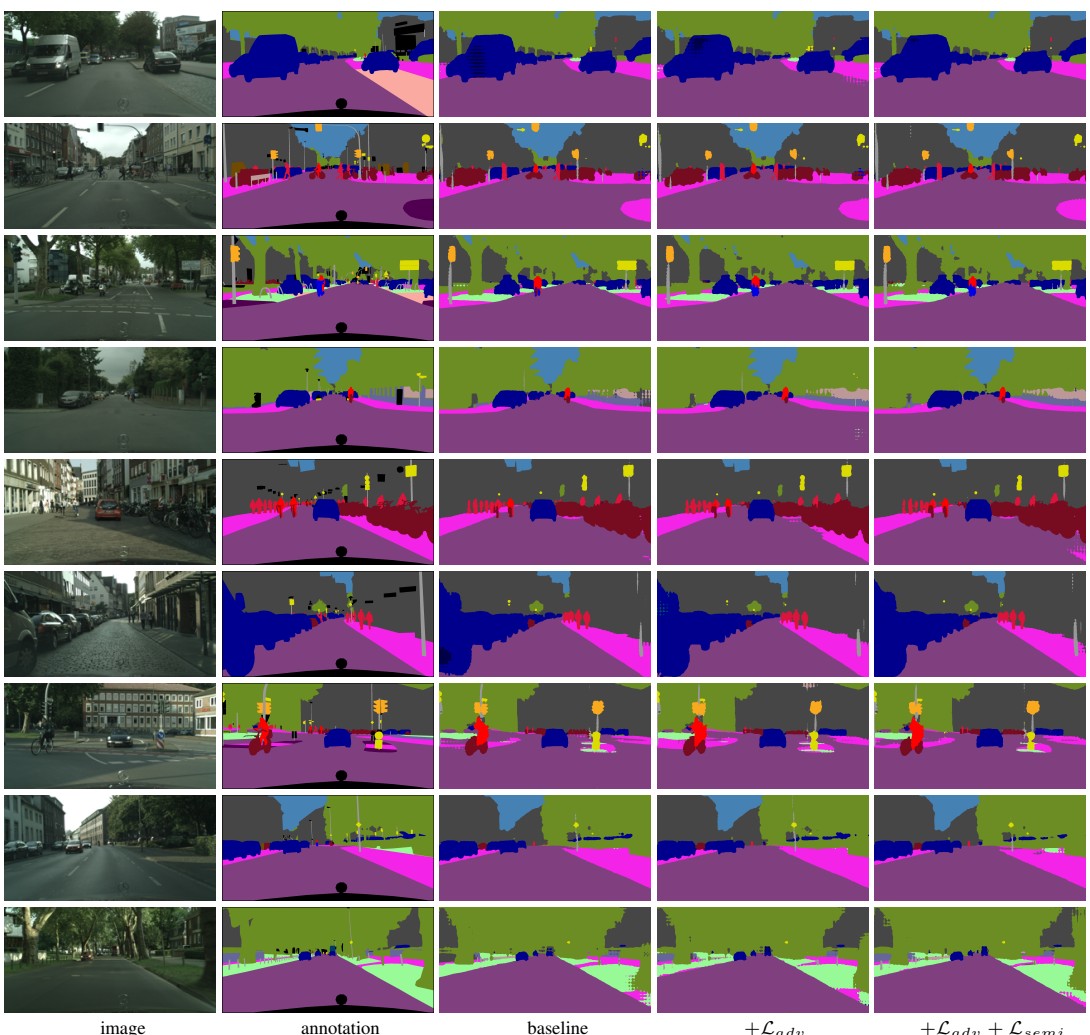

Figure 6: Comparisons on the Cityscapes dataset using 1/2 training data.

