# OpenReview forum: "Adversarial Learning for Semi-Supervised Semantic Segmentation"
_ICLR.cc/2018/Conference — Reject_

### Official Review · AnonReviewer3 · 2017-11-27
**not enough for a first-tier conference**

**Rating:** 5
**Confidence:** 5

**Review:**

This paper describes techniques for training semantic segmentation networks. There are two key ideas:

- Attach a pixel-level GAN loss to the output semantic segmentation map. That is, add a discriminator network that decides whether each pixel in the label map belongs to a real label map or not. Of course, this loss alone is unaware of the input image and would drive the network to produce plausible label maps that have no relation to the input image. An additional cross-entropy loss (the standard semantic segmentation loss) is used to tie the network to the input and the ground-truth label map, when available.

- Additional unlabeled data is utilized by using a trained semantic segmentation network to produce a label map with associated confidences; high-confidence pixels are used as ground-truth labels and are fed back to the network as training data.

The paper is fine and the work is competently done, but the experimental results never quite come together. The technical development isn’t surprising and doesn’t have much to teach researchers working in the area. Given that the technical novelty is rather light and the experimental benefits are not quite there, I cannot recommend the paper for publication in a first-tier conference.

Some more detailed comments:

1. The GAN and the semi-supervised training scheme appear to be largely independent. The GAN can be applied without any unlabeled data, for example. The paper generally appears to present two largely independent ideas. This is fine, except they don’t convincingly pan out in experiments.

2. The biggest issue is that the experimental results do not convincingly indicate that the presented ideas are useful.
2a. In the “Full” condition, the presented approach does not come close to the performance of the DeepLab baseline, even though the DeepLab network is used in the presented approach. Perhaps the authors have taken out some components of the DeepLab scheme for these experiments, such as multi-scale processing, but the question then is “Why?”. These components are not illegal, they are not cheating, they are not overly complex and are widely used. If the authors cannot demonstrate an improvement with these components, their ideas are unlikely to be adopted in state-of-the-art semantic systems, which do use these components and are doing fine.
2b. In the 1/8, 1/4, and 1/2 conditions, the performance of the baselines is not quoted. This is wrong. Since the authors are evaluating on the validation sets, there is no reason not to train the baselines on the same amount of labeled data (1/8, 1/4, 1/2) and report the results. The training scripts are widely available and such training of baselines for controlled experiments is commonly done in the literature. The reviewer is left to suspect, with no evidence given to the contrary, that the presented approach does not outperform the DeepLab baseline even in the reduced-data conditions.

A somewhat unflattering view of the work would be that this is another example of throwing a GAN at everything to see if it sticks. In this case, the experiments do not indicate that it did.

---

> ### Author Response · Authors · 2017-12-13
> **Rebuttal**
>
> We thank the comments and address the raised questions below.
>
> Q1. Why do the authors present two largely independent ideas?
>
> The novelty of this work is to incorporate adversarial learning for dense predictions under the semi-supervised setting without image synthesis. The adversarial learning and semi-supervised learning are not independent in our work. Without the successfully trained discriminator network, the proposed semi-supervised learning does not work well. The ablation study in Table 6 shows that without adversarial loss, the discriminator would treat most of the prediction pixels with low confidence of, providing noisy masks and leading to degenerated performance (drops from 68.8% to 65.7%).
>
> Q2. Why don’t the author use the full DeepLab model?
>
> We implement our baseline model based on the DeepLab in PyTorch for the flexibility in training the adversarial network. We did not use the multi-scale mode in the DeepLab due to the memory concern in section 4.2, in which the modern GPU cards such as Nvidia TitanX with 12 GB memory are not affordable to train the network with a proper batch size. Although this issue may be addressed by the accumulated gradient (e.g., iter_size in Caffe), in PyTorch the accumulated gradient implementation still has issues (ref: https://discuss.pytorch.org/t/how-to-implement-accumulated-gradient/3822/12). We have also verified that it does not work in the current PyTorch version.
>
> However, our main point of the paper is to demonstrate the effectiveness of proposed method against our baseline model shown in Table 1 and 2. In fact, our baseline model already performs better than other existing works in Table 3 and 4.

---

### Official Review · AnonReviewer1 · 2017-11-27
**No title**

**Rating:** 5
**Confidence:** 4

**Review:**

This paper proposed an approach for semi-supervised semantic segmentation based on adversarial training. Built upon a popular segmentation network, the paper integrated adversarial loss to incorporate unlabeled examples in training. The outputs from the discriminator are interpreted as indicators for the reliability of label prediction, and used to filter-out non-reliable predictions as augmented training data from unlabeled images.  The proposed method achieved consistent improvement over existing state-of-the-art on two challenging segmentation datasets.

Although the motivation is reasonable and the results are impressive, there are some parts that need more clarification/discussion as described below.

1) Robustness of discriminator output:
The main contribution of the proposed model is exploiting the outputs from the discriminator as the confidence score maps of the predicted segmentation labels. However, the outputs from the discriminator indicate whether its inputs are from ground-truth labels or model predictions, and may not be directly related to ‘correctness’ of the label prediction. For instance, it may prefer per-pixel score vectors closed to one-hot encoded vectors. More thorough analysis/discussions are required to show how outputs from discriminator are correlated with the correctness of label prediction.

2) Design of discriminator
I wonder if conditional discriminator fits better for the task. i.e. D(X,P) instead of D(P). It may prevent the model generating label prediction P non-relevant to input X by adversarial training, and makes the score prediction from the discriminator more meaningful. Some ablation study or discussions would be helpful.

3) Presentations
There are several abused notations; notations for the ground-truth label P and the prediction from the generator S(X) should be clearly separated in Eq. (1) and (4). Also, it would better to find a better notation for the outputs from D instead of D^(*,0) and D^(*,1).
Training details in semi-supervised learning would be helpful. For instance, the proposed semi-supervised learning strategy based on Eq. (5) may be suffered by noise outputs from the discriminator in early training stages. I wonder how authors resolved the issues (e.g. training the generator and discriminator are with the labeled example first and extending it to training with unlabeled data.)

---

> ### Author Response · Authors · 2017-12-13
> **Rebuttal**
>
> We thank the comments and address the raised questions below.
>
> Q1. How are outputs from discriminator correlated with the correctness of label prediction?
>
> T_semi, # of Selected Pixels (%), Average Pixel Accuracy (%)
> 0,           100%,                                 92.65%
> 0.1,        36%,                                   99.84%
> 0.2,        31%,                                   99.91%
> 0.3,        27%,                                   99.94%
>
> In the above table on the  Cityscapes dataset, we show the average numbers and the average accuracy rates of the selected pixels with different values of T_semi as in (5) of the paper. With a higher T_semi, the discriminator outputs are more confident (similar to ground truth label distributions) and lead to more accurate pixel predictions. Also, as a trade-off, the higher threshold (T_semi), the fewer pixels are selected for back-propagation. This trade-off could also be observed in Table 5 in the paper. We will add more analysis to the paper.
>
> Q2. What’s the performance of D(X,P) compared to D(P)?
>
> We conduct the experiment using D(X,P) instead of D(P) by concatenating the RGB channels with the class probability maps as the input to the discriminator. However, the performance drops to 72.6% on the PASCAL dataset (baseline: 73.6%). We observe that the discriminator loss stays high during the optimizing process and could not produce meaningful gradients. One reason could be that the RGB distributions between real and fake ones are highly similar, and adding this extra input could lead to optimization difficulty for the discriminator network. Therefore, it is reasonable to let the segmentation network consider RGB inputs for segmentation predictions, while the discriminator focuses on distinguishing label distributions. Note that, in Luc2016, similarly the discriminator structure on PASCAL does not include RGB images as inputs. We will add more results and discussions in the paper.
>
> Q3. The notation P in (1) and (4) is not clear.
>
> Thanks for the recommendation. We revise (1) and (4) in the paper for better presentations.
>
> Q4. What are the training details in semi-supervised learning?
>
> We include the details of semi-supervised training algorithm in the revised paper. As the reviewer points out, initial inputs may be noisy, and we tackle this issue by applying the semi-supervised learning after 5k iterations.

---

### Official Review · AnonReviewer2 · 2017-11-29

**Rating:** 5
**Confidence:** 4

**Review:**

The paper presents an alternative adversarial loss function for image segmentation, and an additional loss for unlabeled images.

+ well written
+ good evaluation
+ good performance compared to prior state of art
- technical novelty
- semi-supervised loss does not yield significant improvement
- missing citations and comparisons

The paper is well written, structured, and easy to read.
The experimental section is extensive, and shows a significant improvement over prior state of the art in semi-supervised learning.
Unfortunately, it is unclear what exactly lead to this performance increase. Is it a better baseline model? Is the algorithm tuned better, or is there something fundamentally different compared to prior work (e.g. Luc 2016).

Finally, it would help if the authors could highlight their technical difference compared to prior work. The presented adversarial loss is similar to Luc 2016 and "Image-to-Image Translation with Conditional Adversarial Networks, Isola etal 2017". What is different, and why is it important?
The semi-supervised loss is similar to Pathak 2015a, it would help to highlight the difference, and show experimentally why it matters.

In summary, the authors should highlight the difference to prior work, and show why the proposed changes matter.

---

> ### Author Response · Authors · 2017-12-13
> **Rebuttal**
>
> We thank the comments and address the raised questions below.
>
> Q1. What is the major novelty of this work?
>
> The novelty of this work is to incorporate adversarial learning for dense predictions under the semi-supervised setting without image synthesis. To facilitate the semi-supervised learning, we propose a fully-convolutional discriminator network that provides confident predictions spatially for training the segmentation network, thereby allowing us to better model the uncertainty of unlabeled images in the pixel level. Our model achieves improvement over the baseline model by incorporating this semi-supervised strategy.
>
> Q2. What are the major differences between this work and Luc2016?
>
> The major differences between our work and Luc2016 are listed below:
> - We propose a unified discriminator network structure for various datasets, while Luc2016 designs one network for each dataset.
> - We show that the simplest one-hot encoding of ground truth works well with adversarial learning. The “scale” encoding proposed in Luc2016 does not lead to a performance gain in our experiments.
> - We propose a semi-supervised method coupled with adversarial learning using unlabeled data.
> - We conduct extensive parameter analysis on both adversarial learning and semi-supervised learning, showing that our proposed method performs favorably against Luc2016 with the proper balance between supervised loss, adversarial loss, and semi-supervised loss.
>
> Q3. Differences between this work and Pix2Pix (Isola 2017)?
>
> Our discriminator network works on probability space, while Pix2Pix and other GAN works are on the RGB space. In addition, the target task of Pix2Pix is image translation, and ours is semantic segmentation.
>
> Q4. Difference between this work and constrained CNN (Pathak 2015a)?
>
> In Constrained CNN (CCNN), the setting is weak supervision where image labels are required during training. In our work, we use completely unlabeled images in a semi-supervised setting. Thus, the constraints used by CCNN are not applicable to our scenario where image labels are not available.
>
> In CCNN, they design a series of linear constraints on the label maps, such as those on the segment size and foreground/background ratio, to iteratively re-train the segmentation network. Our framework is more general than CCNN in the sense that we do not impose any hand-designed constraints that need careful designs for specific datasets. Take the Cityscapes dataset as an example, the fg/bg constraint in CCNN does not work in this dataset since there is no explicit background label. The minimum segment size constraint does not make sense either, especially for thin and small objects that frequently appear in road scenes. In contrast, we propose a discriminator with adversarial learning to automatically generate the confident maps, thereby providing useful information to train the segmentation network using unlabeled data.

---

### Public Comment · ~Mohit_Sharma2 · 2017-11-17
**Segmentation Network details**

Thanks a lot for your work. I was trying to reproduce the results of your submission as part of the Reproducibility Challenge. For the baseline model, I have achieved a 52 % mIoU so far. I would like to clarify a few details that might be helpful in replicating the results:

1> What method have you used during the training for upsampling the output map of the DeepLab-v2 network to size 321x321 (input image size for training in PASCALVOC). Currently, I have 3 ConvTranspose2D layers (corresponding to each downsampling layer in the DeepLap-v2 network), each upsampling by a factor of 2.

2> Did you use any other common data preprocessing (like Normalization to 0 mean and 1 variance) ?

Is there any other significant detail that would be helpful in improving the results to match those in the paper?

Thanks again for your work.

---

> ### Author Response · Authors · 2017-11-18
> **Segmentation Network details**
>
> Hi Mohit,
>
> Thanks for interesting in our work. Here are some details that can help yo reproduce our baseline:
>
> 1. Upsampling module: We use 2D bilinear upsampling in our segmentation model (essentially nn.upsample in PyTorch). We use one upsampling module with 8x instead of using three 2x layers. In your case, it would be equivalent to 3 ConvTranspose2D layers with their coefficients initialized as the bilinear kernel with zero learning rate. Intuitively, having upsampling layers with learnable parameters might have better performance due to larger model capacity. But in both our experiments and the original FCN paper from Long et al., learning upsampling does not show significant improvement but introduce much computational overhead in training process.
>
> 2. As mentioned in the paper, we use the Resnet-101 model that is pretrained on the ImageNet. We use the mean and variance for data normalization as the same during pretraining. If you choose to use the torchvision models from PyTorch, the standard data processing transforms are listed in their official docs.
>
> We wish the information can help you in your experiments. Let us know if you encounter any issue. Good luck on the challenge!

---

> > ### Public Comment · ~Mohit_Sharma2 · 2017-11-18
> > **Segmentation Network details**
> >
> > Thanks for your reply. I'll get back to you if I need more help with the experiments.

---

> > > ### Public Comment · ~Mohit_Sharma2 · 2017-11-20
> > > **Adversarial Training Setup**
> > >
> > > Based on your suggestions, I changed my upsampling layers from learnable transposed convolution to simple bilinear upsampling and achieved a mIoU of 69.78. ( As far as I know, now the only difference I have from your submission is using MS COCO pre-trained weights for segmentation network instead of Imagenet. I think I have good enough baseline to continue to the adversarial and semi-supervised training and see if I get a boost by incorporating them on top of my current baseline.) I feel that, because the choice of the upsampling method was so critical in achieving the reported performance of the segmentation network, it would be really helpful if this detail is included in the paper. Anyways, thanks again for giving out the details.
> > >
> > > I would like to ask a few things about the adversarial training used in the paper.
> > >
> > > 1> What scheme did you use for the adversarial-training?
> > > My current idea is something along this line: Take a minibatch of the training set. Perform one forward pass of the segmentation network on this minibatch and update the segmentation-network parameters. For discriminator, calculate the discriminator loss on the class-probability map produced by the segmentation network for the current mini-batch. Then, calculate the discriminator loss on the ground-truth label for the same minibatch. Aggregate the two loss (sum or mean?) and update the discriminator parameters.
> > >
> > > 2> I am not sure about the parameters for the discriminator optimizer. Did you use Nesterov acceleration with Adam? What is the weight decay used (same as generator?)? (I only have a superficial understanding of Adam optimizer. So, I might be missing something obvious.  )
> > >
> > > Thanks.

---

> > > > ### Author Response · Authors · 2017-11-23
> > > > **Adversarial Training Setup**
> > > >
> > > > Hi Mohit,
> > > >
> > > > Thanks for the suggestion. We will add the upsampling details in the following revision. For your information, we will release the source code after the review process.
> > > >
> > > > Regarding your questions:
> > > >
> > > > 1. Yes, we think the way you are implementing it is the same to ours.
> > > >
> > > > 2. Yes, the weight decay/momentum of the discriminator are the same with the generator.
> > > >
> > > > Thanks.

---

> > > > > ### Public Comment · ~Mohit_Sharma2 · 2017-11-30
> > > > > **Adversarial Training**
> > > > >
> > > > > Thanks for your comments.
> > > > >
> > > > >  I was working on stabilizing the GAN training. I couldn't reproduce a significant improvement in mIoU by incorporating adversarial training. I was only able to go up from 68.86% to 68.96% for one of the baseline model. From my side, I have tried to include all the details from the paper.
> > > > >
> > > > > This is my training scheme if you want to have a look. https://gist.github.com/mohitsharma916/c950864e68f719d69a4fbcae3077cf8f
> > > > >
> > > > > and the complete implementation is here
> > > > > https://github.com/mohitsharma916/Adversarial-Semisupervised-Semantic-Segmentation
> > > > >
> > > > > In the meanwhile, I will move on to the semi-supervised training.
> > > > >
> > > > > Looking forward to getting my hands on your implementation to see what I missed. Thanks again for your work.

---

> > > > > > ### Author Response · Authors · 2017-11-30
> > > > > > **Adversarial Training**
> > > > > >
> > > > > > Hi Mohit,
> > > > > >
> > > > > > I found an issue with your implementation. When generating the probability maps, we use SoftMax() instead of LogSoftmax(). If you use LogSoftmax(), the output range will not be 0-1, and the discriminator could easily judge whether the input comes from ground truth or prediction. You can observe the loss of the discriminator whether it is stabilized or not. In our case, the discriminator loss ranges from 0.2-0.4 throughout the training process.

---

> > > > > > > ### Public Comment · ~Mohit_Sharma2 · 2017-11-30
> > > > > > > **Adversarial Training**
> > > > > > >
> > > > > > > Oh! That makes total sense. Thanks a lot for taking time to go through my code.
> > > > > > > I will make the change.
> > > > > > >
> > > > > > > Also, did you use any strategies like, one-sided label smoothing, label flipping etc for stabilizing the GAN training? Or it should work with the settings mentioned in the paper?

---

### Decision · Program_Chairs · 2018-01-29
**ICLR 2018 Conference Acceptance Decision**

**Decision:**

Reject

**Comment:**

The paper presents a reasonable idea, probably an improved version of method (combination of GAN and SSL for semantic segmentation) over the existing works. Novelty is not ground-breaking (e.g., discriminator network taking only pixel-labeling predictions, application of self-training for semantic segmentation---each of this component is not highly novel by itself). It looks like a well-engineered model that manages to get a small improvement with a semi-supervised learning setting. However, given that the focus of the paper is on semi-supervised learning, the improvement from the proposed loss (L_semi) is fairly small (0.4-0.8%).